# GUIDED EVOLUTION WITH BINARY DISCRIMINATORS FOR ML PROGRAM SEARCH

## ABSTRACT

How to automatically design better machine learning programs is an open problem within AutoML. While evolution has been a popular tool to search for better ML programs, using learning itself to guide the search has been less successful and less understood on harder problems but has the promise to dramatically increase the speed and final performance of the optimization process. We propose guiding evolution with a binary discriminator, trained online to distinguish which program is better given a pair of programs. The discriminator selects better programs without having to perform a costly evaluation and thus speed up the convergence of evolution. Our method can encode a wide variety of ML components including symbolic optimizers, neural architectures, RL loss functions, and symbolic regression equations with the same directed acyclic graph representation. By combining this representation with modern GNNs and an adaptive mutation strategy, we demonstrate our method can speed up evolution across a set of diverse problems including a 3.7x speedup on the symbolic search for ML optimizers and a 4x speedup for RL loss functions.

## 1 INTRODUCTION

While neural architecture search (NAS Elsken et al. (2019)) is a major area in AutoML (He et al., 2021), there is a growing body of work that searches for general ML program components beyond architectures, such as the entire learning program, RL loss functions, and ML optimizers (Real et al., 2020; Co-Reyes et al., 2021; Chen et al., 2023b). The underlying search spaces commonly use fine-grained primitive operators such as tensor arithmetic and have few human imposed priors, resulting in long programs or computation graphs with many nodes. While expressive enough to allow discoveries of novel ML components that achieve state-of-the-art results compared to human designed ones, such search spaces have near infinite size. The extremely sparse distribution of good candidates in the space (since small changes usually lead to dramatic performance degradation) poses great challenges for efficient search of performant programs. Unlike NAS where several successful search paradigms exist (Bayesian optimization (Kandasamy et al., 2018; White et al., 2021a), reinforcement learning (Zoph & Le, 2017; Pham et al., 2018), differentiable search (Liu et al., 2018b)), regularized evolution (Real et al., 2019) remains a dominant search method on these primitive search spaces due to its simplicity and effectiveness. The main technique for speeding up search on these spaces so far is functional equivalent caching (FEC Gillard et al. (2023)), which skips repeated evaluation of duplicated candidates. While effective, FEC does not exploit any learned structure in the evaluated candidates. Can we devise learning-based methods that capture global knowledge of all programs seen so far to improve search efficiency?

Performance predictors (White et al., 2021b) have shown successes in speeding up search in many NAS search spaces but have not yet been tried on these larger primitive-based search spaces. Prior work uses regression models, trained to predict architecture performance, to rank top candidates before computationally expensive evaluation. An alternative is to train binary relation predictors (Dudziak et al., 2020; Hao et al., 2020), which predict which candidate from a pair is better. Previous work usually trains performance predictors on NAS search spaces using only a few hundred (or fewer) randomly sampled candidates (Dudziak et al., 2020). Prediction on unseen candidates relies on strong generalization performance and we argue that this is relatively easy for many NAS search spaces, but much harder for these primitive search spaces. Ying et al. (2019) showed the NAS-Bench-101

fitness landscape is very smooth with most candidates having similar fitness and the global optimum is at most 6 mutation steps away from more than 30% of the search space. Li & Talwalkar (2019) further showed that random search is a strong baseline on NAS-Bench-101 which suggests that a predictor trained on a fixed offline dataset collected from random sampling will generalize well. In contrast, in Real et al. (2020) and Chen et al. (2023b), random search fails completely on these larger search spaces which have sparse reward distribution. So random search alone will struggle to capture enough representative data for generalization.

In this paper, we propose training a binary relation predictor online to guide mutations and speed up evolution. This predictor is trained continuously with pairs of candidates discovered by evolution to predict which candidate is better. We introduce a novel mutation algorithm for combining these predictors with evolution to continually score mutations until we find a candidate with a higher predicted fitness than its parent, bypassing wasteful computation on (likely) lower fitness candidates. Our method provides large benefits to evolution in terms of converging more quickly and to a higher fitness over a range of problems including a 3.7x speedup on ML optimizers and 4x speedup on RL loss functions. We show that obtaining generalization with this binary predictor is much easier than with a fitness regression model and that using state-of-the-art graph neural networks (GNNs) gives better results. Our unified representation and training architecture can be applied to generic ML components search.

## 2 RELATED WORK

**NAS with performance predictors.** Many performance predictor methods (White et al., 2021b) have been proposed to speed up NAS. Once trained, these models are used for selecting the most promising architecture candidates for full training, reducing the resources (compute and walltime) wasted on unpromising ones. One popular category of these methods is to train a regression model (Liu et al., 2018a; Wen et al., 2020; Lu et al., 2021; Sun et al., 2020; Peng et al., 2022) to predict performance of an architecture solely based on its encoding. Some regression models, such as ReNAS (Xu et al., 2021), are trained directly with a ranking loss instead of MSE. An alternative is to train pairwise binary relation models (Dudziak et al., 2020; Hao et al., 2020; Chen et al., 2021). Such model takes a pair of candidates as input and predicts which candidate is better. This is motivated by the observation that predicting relative performance of a pair of candidates is sufficient for ranking. BRP-NAS (Dudziak et al., 2020) shows that binary predictor models are more effective than regression models for candidate selection. BRP-NAS alternates between two phases: i) use the predictor to rank candidates and ii) select the top candidates for full training and the results are used to improve the predictor. In Dudziak et al. (2020), these two phases alternate very few times to collect no more than a few hundred fully trained candidates. Fewer works explore the integration of binary predictor with evolution. To our best knowledge, Hao et al. (2020) is the closest to our work, which uses the predictor to rank a list of offspring candidates by doing pairwise comparisons between candidates. In contrast, our work uses the binary predictor to compare the child candidate with the parent (the tournament selection winner) which explicitly encourages hill climbing (Russell & Norvig, 2010). In addition, problems studied in this work are of much larger scale compared to those in Hao et al. (2020). We note that hill climbing strategy has been tried for NAS in Elsken et al. (2017), however this does not make use of performance predictors and evolutionary algorithms.

**General ML program component search.** Recent work (Real et al., 2020; Co-Reyes et al., 2021; Chen et al., 2023b) searches for ML program components beyond neural network architectures. These search spaces use primitive operators as building blocks and few human priors are imposed on how they should be combined. This is in stark contrast with many NAS search spaces where non-trivial human priors are included in the design. For example, it is a common choice to bundle ReLU, Convolution, and Batch Normalizaton as one atomic operator (in a fixed order), as in NAS-Bench-101 and DARTS Liu et al. (2018b). Due to the use of primitive operators and few constraints, these general ML program search spaces are much larger and have sparse rewards. This makes them more challenging than typical NAS search spaces. Previously, hashing based techniques (Gillard et al., 2023) are the main method for speeding up search on these large primitive search spaces, while predictive models have not been tried. We show that predictive models can provide extra speedup on top of FEC techniques in the ML optimizer search space.

**Learning and program synthesis.** There are many works that learn a generative model or policy over discrete objects. Work such as Zoph & Le (2017); Ramachandran et al. (2018); Bello et al. (2017); Abolafia et al. (2018); Petersen et al. (2021) use reinforcement learning to optimize this model. Other works combine a generative model with evolution such as using an LLM as a mutator (Chen et al., 2023a) or training a generator to seed the population (Mundhenk et al., 2021). Training a generative model in this large combinatorial space is generally more difficult than training a binary discriminator and requires more complicated algorithms such as RL whereas our method is a simple modification to evolution.

## 3 METHOD

In this section we describe the search representation, how the binary predictor can work over a variety of ML components, and how to combine the model with evolution.

### 3.1 SEARCH REPRESENTATION

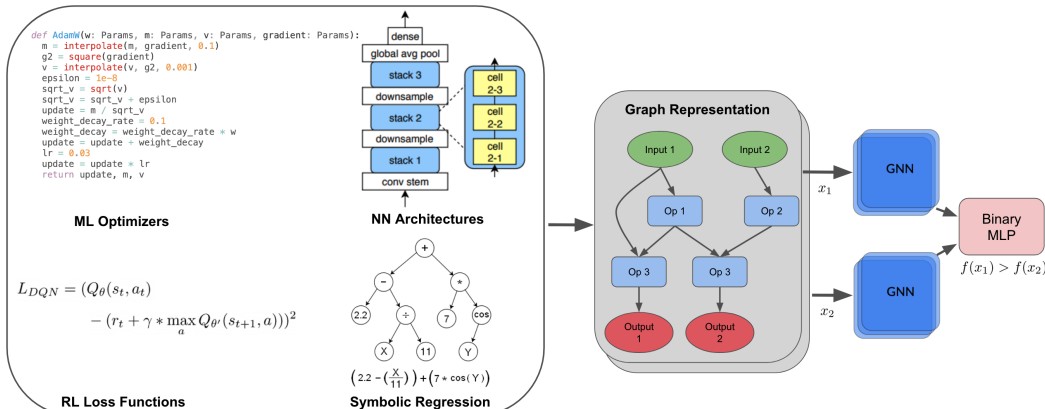

Figure 1: We encode a variety of ML components (learning optimizers input as Python code, NN architectures, RL loss functions, and symbolic equations) into the same computation graph representation and learn a GNN-based binary predictor over pairs of individuals to predict which graph has better performance.

We build a general framework to encode a wide variety of ML components with the same GNN architecture. We first convert each task (as described below) into a directed acyclic graph (DAG) where a DAG consists of input nodes, intermediate operation nodes, and output nodes. A directed edge from node $a$ to node $b$ would indicate that $a$ is an input for operation $b$. The number of possible identities for each node will depend on the possible operations for that task.

**Neural Network Architectures**: A neural network architecture cell is readily represented as a DAG where nodes are operations such as 3x3 convolution and edges define data flow between ops (Ying et al., 2019).

**Symbolic Regression**: The objective is to recover a range of target equations such as $log(1+x)+x^2$. While equations are normally represented as trees, the DAG formulation allows variables to be reused since children can have multiple parents. Node ops are basic math operators Uy et al. (2011).

**ML Optimizer Algorithms**: The optimizer for updating a neural network such as AdamW (Loshchilov & Hutter, 2019) is parsed as Python code Chen et al. (2023b). Each line is an assignment that becomes a node in the DAG similar to Koza (1993). Node ops can be math operators or high level function calls.

**RL Loss Functions**: The loss function to train an RL agent such as DQN (Mnih et al., 2013) is converted into a DAG as in Co-Reyes et al. (2021). Nodes are operations such as apply a Q network on an environment observation and the output node is the final loss to minimize.

## 3.2 ARCHITECTURE + TRAINING

We describe the general architecture and how to train the binary predictor. Once a problem can be converted into a DAG, we compute embeddings for each node and edge based on its identity. Between domains, node operations will differ so we use different embedding layers between tasks where the number of possible embeddings depends on the maximum number of operations. Edge embeddings will similarly depend on the max number of possible connections for that task. Given these embeddings, we can then apply a GNN or graph Transformer to compute the DAG encoding $g(x)$ of any graph $x$.

Given a dataset of individuals, we want to train a predictor $f$, that can take in any pair of individuals $(x_1, x_2)$ and predict whether the fitness of $x_1$ is greater than the fitness of $x_2$. Our predictor is a 2 layer binary MLP which takes in the concatenated graph encodings $concat(g(x_1), g(x_2))$ and outputs a logistic score. The predictor and GNN are trained end-to-end to minimize the binary cross entropy loss over randomly sampled pairs from the current dataset. During evolution, this predictor is trained online and so will improve as the dataset size increases with more individuals. The dataset is a fixed size queue and training happens at set intervals during evolution. More details on the architecture and training algorithm are in Appendix 6.1.

## 3.3 COMBINING BINARY MODELS WITH EVOLUTION

A binary predictor $f(\cdot, \cdot)$ can be combined with a vanilla mutator (e.g. modifying a few nodes and edges in a DAG) to form what we call a *mutation strategy*. There are many possible mutation strategies and we propose a particular design which achieves good performance.

We briefly recap how regularized evolution works and introduce a few terms. Regularized Evolution (RegEvo) has two phases: In the first phase, we initialize a *population* queue of size $P$ with randomly generated candidates, each candidates is assigned a *fitness* score. In the second phase, we repeat the following loop until we have swept over a desired number of candidates: i) Randomly sample $T$ candidates from $P$ and select the one with the highest fitness. This step is known as *tournament selection* (Goldberg & Deb, 1990) of size $T$; ii) A *mutator* randomly mutates the selected candidate $p$ to a child $c$, and $c$'s fitness score is computed; iii) Add $c$ to the end of the queue and remove the oldest item from the head of the queue.

---

**Algorithm 1** Binary Predictor-based Adaptive Mutation with Re-Tournament (PAM-RT)

---

**Input:** Population buffer $P$, trained predictor $f$, max attempts K.

1: $accept \leftarrow False$
2: $attempts \leftarrow 0$
3: **while** $accept == False$ and $attempts < K$ **do**
4:     $parent \leftarrow tournament\_selection(P)$
5:     $child \leftarrow mutate(parent)$
6:     $accept \leftarrow f(child, parent) > 0.5$
7:     $attempts \leftarrow attempts + 1$
8: **end while**
**Output:** $child$

---

In prior work combining predictors with evolution, it is common to generate a list of candidates and use the predictor to rank these candidates. For example, in Hao et al. (2020) a parent is mutated to a list of child candidates, and the model is used to score each candidate (against other candidates) and take the argmax as the final candidate. More precisely, if we have a list of candidates $c_i$, then $score(c_i) = \sum_{j \neq i} f(c_i, c_j)$ and $f$ takes discrete values in $\{-1, 1\}$. We call this approach max pairwise (**Max-Pairwise**).

In this work, we explore a different family of strategies based on the heuristic of hill climbing (Russell & Norvig, 2010) which performs iterative local search to find incrementally better solutions. We use the binary model to compare the child against the parent (instead of against each other), and this explicitly encourages selecting a child that is likely better than the parent. In its most basic form, we run tournament selection once to obtain a parent $p$, mutate it to obtain a child candidate $c$, and check if the child is likely better than the parent by evaluating $f(c, p)$; if not, we retry the mutation from $p$ and otherwise accept $c$. We refer this method as predictor-based adaptive mutation (**PAM**). A variant is to retry the tournament if the mutated child is rejected by the model which we refer to as predictor-based adaptive mutation with re-tournament (**PAM-RT**). We summarize PAM-RT in Algorithm 1. PAM can be obtained by a one line change by moving the tournament-selection before the while-loop.

**Hill climbing properties of PAM and PAM-RT.** We provide some motivation for PAM-RT. Given a standard evolution mutator $m$, the *natural hill climbing rate* $q$ at a point $p$ is the probability that the random child $m(p)$ is better than $p$. We define the *modified hill climbing rate* as the same probability but for a child produced from our proposed method. PAM-RT has an extra re-tournament mechanism compared to PAM which gives PAM-RT a bias towards selecting parents that are more likely leading to a hill climbing child – *if the model is better than random*. At each iteration in PAM-RT, the probability of accepting a child is $p_{accept} = (2a-1)q + (1-a)$ (using Eq. 1 in Appendix 6.2), which increases with the hill climbing rate $q$ if the model accuracy $a > 0.5$. Therefore PAM-RT will accept children of parents with higher hill climbing probability more quickly on average.

## 4 EXPERIMENTS

In our experiments we show that PAM-RT can provide a meaningful speedup to evolution. We perform ablations showing the importance of several design choices and provide analysis on how to combine our predictor with evolution.

### 4.1 EXPERIMENTAL SETUP

We describe the search spaces, their fitness definitions, and the random generators and mutators we use in regularized evolution.

**NAS-Bench-101** Ying et al. (2019) is a NAS benchmark for image classification on CIFAR-10 (Krizhevsky, 2009). The search space consists of directed acyclic graphs (DAGs) with up to 7 vertexes and up to 9 edges. Two special vertexes are `IN` and `OUT`, representing the input and output tensors of the architecture. Remaining vertexes can choose from one of three operations: 3x3 convolution, 1x1 convolution, and 3x3 max-pool. This search space contains 510M distinct architectures (of which 423K are unique after deduping by graph isomorphisms). Every candidate in the search space has been pre-evaluated with metrics including validation accuracy and test accuracy. Evaluation of a single candidate is a fast in-memory table lookup which enables noisy oracle experiments in Section 4.2.2. In this work, we use the validation accuracy (after 108 epochs of training) as the fitness. We use the same random generator and mutator as published in Ying et al. (2019). Although NAS-Bench-101 does not use a primitive search space, we include it here because it is a well-accepted benchmark supporting extensive ablation studies. It is also a test to see if our proposed method works on more traditional NAS spaces.

**Nguyen** Uy et al. (2011) is a benchmark for symbolic regression tasks. The goal is to recover ground truth single-variable or two-variable functions. Candidate functions are constructed from $\{+, -, *, /, \sin, \cos, \exp, \log, x\}$ (and additionally with $\{y\}$ for two-variable functions). We choose the Ngyuen benchmark because it shares features with primitive-based AutoML search spaces, such as having sparsely distributed high performing candidates. We choose a few Nguyen tasks (Nguyen-2, Nguyen-3, Nguyen-5, Nguyen-7, Nguyen-12) to represent varied search space sizes and difficulties. Given a candidate, we first compute the root mean square error (RMSE) between the candidate and target function over a set of uniformly sampled points from the specified domain of each task (e.g. 20 points in $(0, 1)$ for Nguyen-12). The fitness is obtained by applying a "flip-and-squash" function $(2/\pi)\arctan(x\pi/2)$ so that fitness is within $[0, 1]$ and lower RMSE maps to higher fitness. If any candidate function generates NAN, its fitness is defined as 0. The DAG has a maximum of 15 vertices with 8 possible ops.

**AutoRL Loss Functions** introduced in Co-Reyes et al. (2021) are represented as DAGs and the goal is to find RL loss functions that enable efficient RL training. The DAG has a maximum of 20 vertexes excluding the input and parameters vertexes. Remaining vertexes can choose from a list of 26 possible ops as detailed in Co-Reyes et al. (2021). We use three environments (CartPole-v0, MountainCar-v0, Acrobot-v1) from the OpenAI Gym suite Brockman et al. (2016) with the CartPole environment used as a "hurdle" environment. Fitness is defined as the average normalized return from all three environments if the performance is greater than 0.6 on Cartpole, otherwise only the normalized return from Cartpole is used (and the rest two environments are skipped to save compute). Random generators and mutators are the same as in Co-Reyes et al. (2021).

**ML optimizers (Hero)** Chen et al. (2023b) defines a search space for ML optimizers and uses evolution to discover a novel optimizer that achieves state-of-the-art performance on a wide range of

machine learning tasks. The search space consists of a sequence of python assignment statements which can use common math functions or high level functions such as linear interpolation (example in Figure 1 top left). We convert the sequence of statements to an equivalent DAG. Evolution is warm started from AdamW and fitness is the validation log likelihood loss of the trained model. Chen et al. (2023b) uses up to 100 TPUs v4 per experiment so to reduce compute, we focus on a smaller training configuration that can train an optimizer on a language modelling task in less than 10 minutes on a GPU (Nvidia Tesla V100).

**Evolution and trainer setup**

For population and tournament sizes, (P, T), we use (100, 20) for NAS-Bench-101, (100, 25) for Nguyen, (300, 25) for AutoRL, and (100, 25) for Hero. We swept over tournament size for the baseline. For all tasks we use the GPS graph Transformer (Rampásek et al., 2022) except for ML optimizers where we use SAT Chen et al. (2022) for the graph encoder. Samples from evolution are added to a fixed size replay buffer that is used to train the model online (Algorithm 2). We use Adam with a learning rate of $1e-4$ and train the model over the replay buffer at a fixed frequency as new samples are added. More details on architecture and training are in Appendix 6.1. For evolution curves, we plot $95\%$ confidence intervals ($\pm 2$ standard errors) over at least 5 seeds.

## 4.2 RESULTS

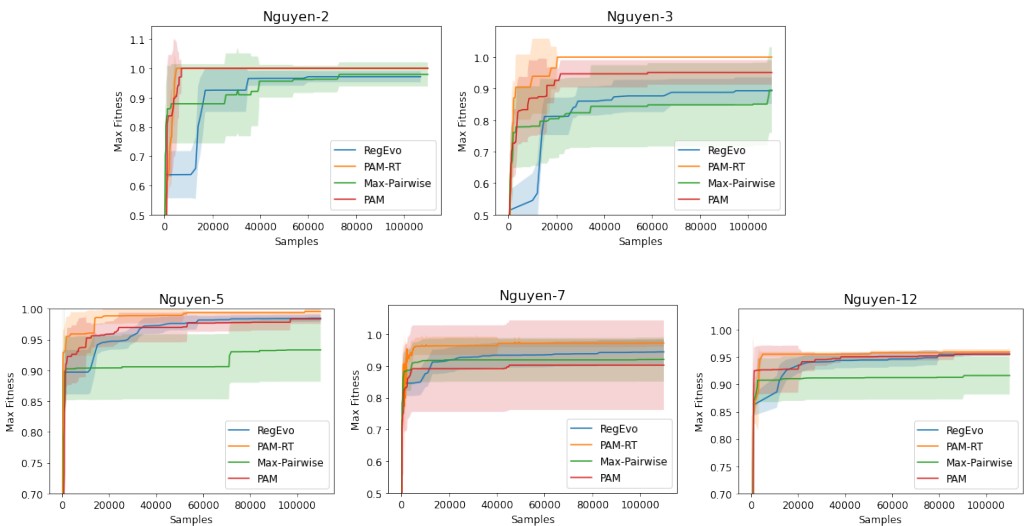

Figure 2: On all symbolic regression tasks, our method PAM-RT can provide faster convergence compared to regularized evolution. PAM-RT also outperforms other mutation strategies, Max-Pairwise and PAM.

### 4.2.1 TRAINED PREDICTORS SPEED UP EVOLUTION

We perform experiments on larger primitive-based search spaces to see if our method can speed up evolution. Nguyen, AutoRL, and Hero have much larger and sparser search spaces than NAS-Bench-101. Search space size can approximately be measured by number of vertices and possible ops for each vertex. These 3 task for (# nodes, # ops) are (15, 8), (20, 26), and (30, 43) whereas NAS-Bench-101 is (7, 3). In Figure 2 and 3, we show that our method (PAM-RT) increases the convergence speed of evolution and reaches a higher maximum fitness with fewer samples compared to regularized evolution. Across all 5 Nguyen tasks, our method significantly speeds up evolution and for example achieves the maximum fitness on Nguyen-3 in 20k samples while RegEvo fails to reach the maximum fitness in 100k samples. On Hero, our method achieves the same average maximum fitness in 10k samples as RegEvo does in 37k samples (both the baseline and PAM-RT use FEC). On AutoRL, our method achieves the same fitness as RegEvo in 4x less compute.

Comparing mutation strategies, we see that PAM-RT generally outperforms PAM and Max-Pairwise. While Max-Pairwise can sometimes obtain good early performance, it converges to a lower maximum fitness later on. Max-Pairwise chooses the highest scoring candidate so it could be exploiting the model too heavily, leading to less exploration.

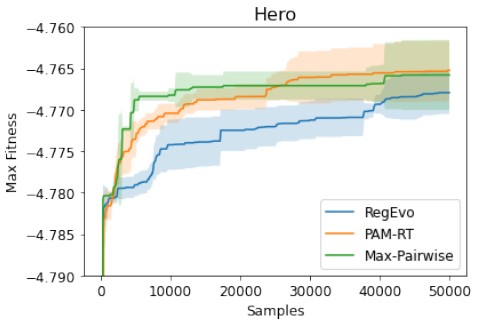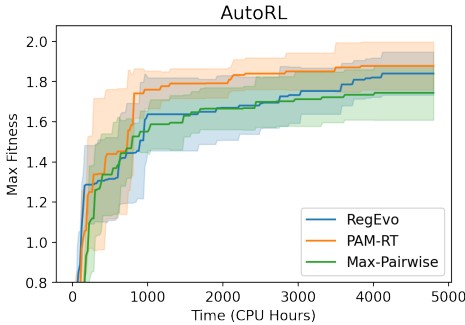

Figure 3: PAM-RT has better sample efficiency and higher maximum performance compared to regularized evolution on harder search spaces, Hero and AutoRL.

### 4.2.2 EFFECT OF MODEL ACCURACY ON FINAL PERFORMANCE

Although a good predictor can improve local search, it is not guaranteed that it improves long term performance even if the model is perfect. In addition, training an accurate predictor online confounds with the predictor's effect on evolution. For example, an inaccurate model may lead to the collection of more sub-optimal data for training, creating a negative feedback loop. We measure the effect of a model's simulated accuracy on evolution by assuming access to an oracle model and adjusting its accuracy. We conduct (noisy) oracle experiments using NAS-Bench-101 and Nguyen tasks because oracle models are available for them (pre-computed for NAS-Bench-101 and fast to compute for Nguyen tasks). Given an oracle $g(\cdot)$ that assigns a fitness for any candidate, we can simulate a binary prediction model $f(\cdot, \cdot)$ of any accuracy $a$ by randomly flipping the ground truth ordering with probability $1 - a$. We use the noisy oracle models in the PAM-RT setting. Figure 4 (left)

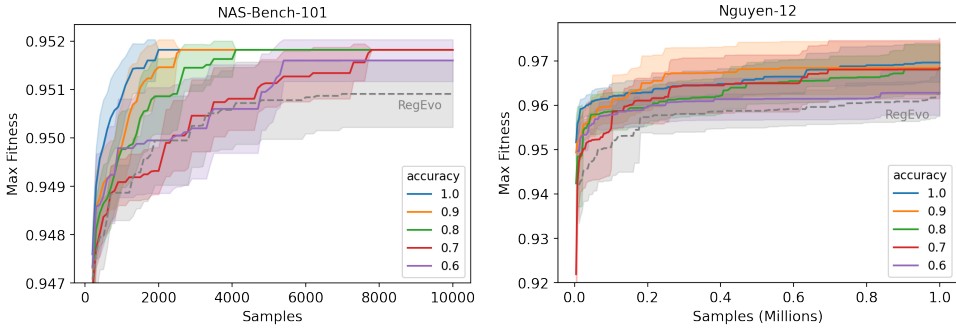

Figure 4: Noisy oracle experiments on NAS-Bench-101 and Nguyen-12 show the benefit of a using a predictor with evolution. Dashed curves show regularized evolution baseline.

shows that on NAS-Bench-101 evolution performance correlates well with model's accuracy where a perfect model ($100\%$ accuracy) converges the fastest and the least accurate model ($60\%$) converges the slowest. These experiments highlight the importance of the predictor's accuracy for end-to-end performance, motivating us doing design ablations in Section 4.2.3 to identify models that work best.

### 4.2.3 DESIGN ABLATIONS

Here we show what design choices matter for PAM-RT. For ablations, we use a set of Nguyen tasks as they have varying difficulty and search space sizes while still having cheap evaluation for easier analysis.

**Binary vs Regression:** Most prior work uses a regression predictor so we study the effect of using a regression vs a binary predictor. We collect $10,000$ samples from regularized evolution, train for $1000$ epochs on a training set, and measure accuracy on a held out test set. The accuracy for the regression predictor is measured by comparing which of the two predicted scores is higher for a pair of graphs. In Figure 5, we show that the binary predictor achieves significantly higher test accuracy across all symbolic regression tasks. This further motivates the use of binary predictors since accuracy can have a large impact on evolution convergence as shown in Figure 4 (left). As Dudziak et al. (2020) pointed out, a binary predictor can leverage $O(N^2)$ training samples with $N$ evaluations, but a regression can only use $O(N)$ training samples. A regression predictor must also generalize to unseen higher fitnesses while a binary predictor just has to predict comparisons. This partly explains why they are more effective and generalize better.

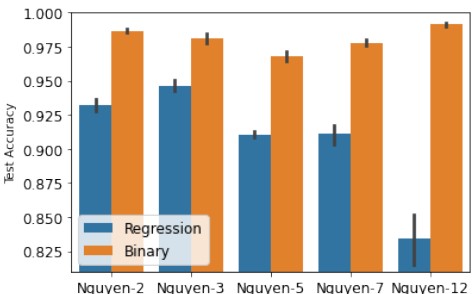
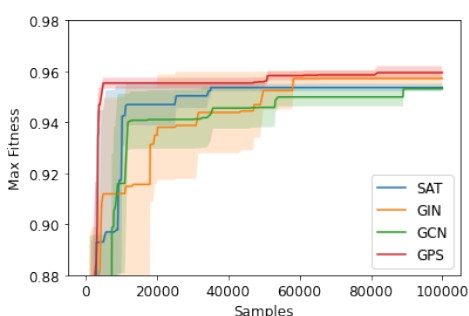

Figure 5: **Left**: Test accuracy for regression vs binary predictor on random pairs from a fixed 10k dataset collected with regularized evolution for a range of symbolic regression tasks. Binary predictors have consistently higher test accuracy. **Right**: Task performance on Nguyen-12 task using different GNN architecture with GPS being the most performant architecture.

**Does the choice of GNN matter?** We perform an ablation experiment comparing different popular GNNs and graph Transformers including GPS Rampásek et al. (2022), SAT Chen et al. (2022), GCN Kipf & Welling (2017), and GIN Xu et al. (2019). We observe that the choice of GNN architecture has an effect on the convergence of evolution. For the hardest symbolic regression task, Nguyen-12, GPS performs the best in terms of converging quickly to the highest max fitness. Older networks such as GCN and GIN take longer to converge and reach a lower max fitness.

### 4.3 EMPIRICAL ANALYSIS

To better understand how the model helps and where it fails, we perform a counterfactual experiment where we run regularized evolution as usual and train the PAM-RT model online but do not use it for mutations.

This gives insight into the performance of the model given the ground truth. We run this experiment on Nguyen-5 for easier analysis. At each mutation step, we sample 64 candidates, use the model to score these candidates, and save the score and actual fitness of each candidate. We run this for 100k steps to get $6.4e6$ scores. We define a positive individual to be the case where its fitness is higher than the parent, and a negative individual is defined as the complement. In Figure 6, we plot the accuracy curve and the precision-recall curve of these candidates over a range of classifier thresholds. As expected, accuracy is relatively high ($> 0.95$ for a classifier threshold of $0.5$). During evolution most individuals are negatives and the model correctly scores most of these individuals with low scores. We observe this in Figure 7 where most of the negative scores are less than $0.5$. However, in Figure 6, we see that precision is quite low (around $0.01$) and drops quickly to close to $0$ for higher levels of recall. Positive individuals are quite rare during evolution and while the model can correctly score some of them ($0.43$ true positive rate for $0.5$ threshold) as seen in Figure 7, there are a good portion of positives with less than $0.5$ score. From this analysis, we see that the model is good at ruling

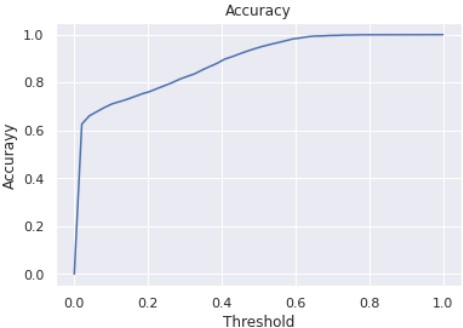 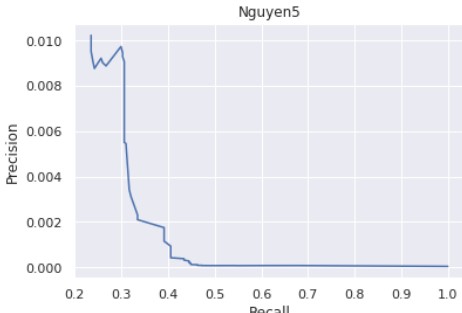

Figure 6: **Left**: Accuracy of our model for various classifier thresholds. Evaluated on data collected during RegEvo with a model trained online on Nguyen5. **Right**: Precision recall curve of our model.

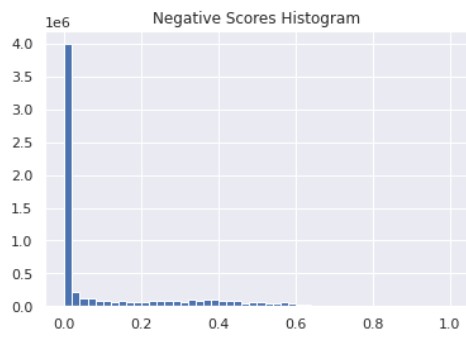 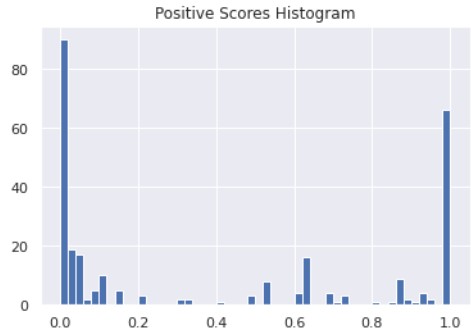

Figure 7: **Left**: Histogram of scores for negative individuals. **Right**: Histogram of scores for positive individuals.

out many negative individuals and this would logically make evolution more efficient. However the model could be better at letting more positives individuals through and reducing the number of false negatives. Further analysis of why there is a high number of false negatives with almost 0 score could provide insight for how to improve the model.

## 5  DISCUSSION

We have presented a method for speeding up evolution with learned binary discriminators to more efficiently search for a wide range of ML components. The same graph representation and GNN-based predictor can be used over diverse domains and provide faster evolutionary search for symbolic reqression equations, ML optimizers, and RL loss functions, as well as traditional NAS spaces. Through ablations, we showed the importance of the mutation strategy, the use of a binary predictor instead of a regression model, and state-of-the-art GNN architectures.

Predictor's accuracy and generalization capability are important for this method to provide large benefits. Immediate future work could focus on representation learning for better generalization over graphs such as better graph architecture priors, unsupervised objectives, or even sharing data across tasks. Longer term, one could consider alternative optimization methods for searching over ML programs, but that still use learned representations over computation graphs such as an RL policy. One potential avenue could be using generative models such as LLMs to propose promising candidates. Speeding up the search for ML components with learning is a promising direction because it could eventually create a virtuous cycle of continuous self improvement.

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
