# 6 SUPPLEMENTARY MATERIAL

## 6.1 TRAINING DETAILS

**Single-process training**. For smaller scale tasks (NAS-Bench-101 and Ngyuen), we use a single-process training procedure (Algorithm 2) which alternates between two phases: collecting new samples and fitting the binary predictor. An optional exploration parameter $\epsilon$ may be used to allow a small probability of using the vanilla mutator instead of a mutation strategy with the predictor.

**Distributed training**. For larger scale tasks (Hero and AutoRL), we use asynchronous distributed training to speed up candidate evaluation and model training. We use a central *population server* that tracks all the candidates discovered so far while maintaining a population buffer. A single *learner* periodically syncs new candidates data from the population server and maintains the bounded replay buffer. The learner starts learning as soon as enough data has been collected, and it learns continuously—not blocked by workers' progress. Parallel *worker*s sync population buffer and model parameters from the population server and the learner respectively. Workers are responsible for tournament selection, mutation, and candidate evaluation. Candidates are sent to the population server once evaluation completes.

**Architecture details**. For the graph encoder, we use the standard GPS architecture from Pytorch Geometric Fey & Lenssen (2019) with 10 layers of GPS convolutions. Nodes and edges from the DAG are input into an embedding layer of size 64 and 16 respectively to obtain node and edge embeddings which are then processed by the GNN. The GPS convolution has 64 input channels, 4 heads with an attention dropout of 0.5, and uses a GINE Hu et al. (2019) message passing layer which uses a 2 layer (64, 64) MLP. A final global addition pooling layer across node features is applied to obtain the graph embedding of size 64. The binary predictor is a 2 layer MLP of size (64, 64) with dropout of 0.2. For all MLPs, we use ReLU activations.

**Other training details**. The predictor is trained with Adam with a learning rate of $1e-4$ and weight decay of $1e-5$. Samples from evolution are added to the replay buffer. For NAS-Bench-101 we train the predictor for 100 epochs every 100 samples. For Nguyen, we train the predictor for 10 epochs every 100 samples. One epoch is iterating over the shuffled replay buffer twice to obtain pairs of graphs and then minimizing the binary classification loss on these pairs. For Hero and AutoRL, we use the asynchronous distributed training logic described above. We detail configurations and hyperparameters in Table 1.

| Task | NAS-Bench-101 | Nguyen | Hero | AutoRL |
|---|---|---|---|---|
| Population Size | 100 | 100 | 100 | 300 |
| Tournament Size | 20 | 25 | 25 | 25 |
| Num Workers | 1 CPU | 1 GPU | 20 GPUs | 100 CPUs |
| Replay Buffer Size | 1000 | 10000 | 50000 | 100000 |
| Min Data Before Model Use | 100 | 100 | 1000 | 3000 |

Table 1: Hyperparameters for tasks

**Compute budget**. Evaluating time for a candidate varies greatly for AutoRL (from under one minute to over an hour due to the hurdle mechanism). Therefore, we set a timing budget of total 4800 CPU hours (summed from 100 parallel workers), instead of a budget on number of samples, as they vary greatly for a given time budget.

**Training tricks**. We discuss important tricks for training and using the model. We found that delaying using the model until enough samples have filled the replay buffer to be important for Hero (1000 samples) and AutoRL (3, 000 samples). This could be because these search spaces are larger, so the model can easily overfit to a small number of samples and struggle to generalize. Another important hyperparameter related to this one, was using a large enough replay buffer size. Smaller buffer sizes ($< 1000$) had a detrimental effect.

---

**Algorithm 2** Online Training of Binary Predictors with Evolution

---

**Input:** Total samples $S$, random mutation probability $\epsilon$, training frequency $F$, max attempts $K$.
1: Initialize population buffer $P$, predictor $f$, replay buffer $D$
2: $samples \leftarrow 0$
3: **while** $samples < S$ **do**
4:     **if** $Uniform(0,1) < \epsilon$ **or** $samples < $ min data **then**
5:         $child \leftarrow RandomMutation(P)$
6:     **else**
7:         $child \leftarrow$ PAM-RT$(P, f, K)$             ▷ Select child with Algorithm 1
8:     **end if**
9:     **if** $samples \bmod F == 0$ **then**
10:         $f \leftarrow TrainBinary(f, D)$
11:     **end if**
12:     $D \leftarrow D \cup child$                  ▷ Remove oldest if hit max $D$ size
13:     $P \leftarrow P \cup child$                  ▷ Remove oldest if hit max $P$ size
14:     $samples \leftarrow samples + 1$
15: **end while**

---

## 6.2 HOW PAM IMPROVES LOCAL SEARCH

We show how the modified hill climbing rate (defined in Section 3.3) relates to the natural hill climbing rate for a given model accuracy using PAM. As a corollary, we show the modified rate is always higher than the natural rate—*if the model is better than random.*

For simplicity, assume we can retry as many times as needed (i.e., $K \to \infty$ in Algorithm 1). Let $p$ denote the parent, $c$ denote the child, $q$ be the probability $c > p$ and $a$ be the binary predictor's accuracy. At each step, the probability of accepting $c$ is:

$$p_{accept} = qa + (1-q)(1-a). \tag{1}$$

Here we assume the event $c > p$ (or $c \leq p$) and the event that model makes a correct (or incorrect) prediction are independent. Let $d = 1 - p_{accept}$ denote the probability we reject $c$.

For a sequence of trials, let random variable $C_i \in \{0,1\}$ denote if the child sampled at $i$-th round is better than the parent (*according to the ground truth*) and let the random variable $A_i \in \{0,1\}$ denote we accept the child *according to the model*. The probability that we eventually accept a child that's better than the parent is the sum of probability of the following mutually exclusive events:

1. The first mutation is good and we accept it: $P(C_1 = 1)P(A_1 = 1|C_1 = 1) = q \cdot a$.

2. We reject the first mutation, and the second mutation is good and we accept it: $P(A_1 = 0)P(C_2 = 1)P(A_2 = 1|C_2 = 1) = d \cdot q \cdot a$.

3. In general, $d^{n-1} \cdot q \cdot a$ if we accept at $n$-th trial ($n$ starts with 1).

Summing over the geometric series gives

$$
\begin{aligned}
q \cdot a \cdot \frac{1}{1-d} &= \frac{qa}{p_{accept}} \\
&= \frac{qa}{qa + (1-q)(1-a)} \\
&= \frac{1}{1 + \frac{1-a}{a} \cdot (\frac{1}{q} - 1)},
\end{aligned}
$$

recovering the result in Section 3.3. It is easy to see that if the model accuracy is better than random ($a > 0.5$), the modified hill climb rate is always higher than the natural hill climb rate. Increasing model accuracy leads to larger gains (Figure 8).

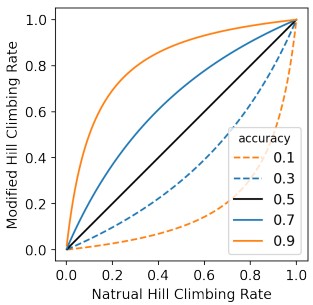

Figure 8: Modified hill climbing rate as a function of natural hill climbing rate for different model accuracy levels (using PAM).

## 6.3 EXPLOITATION AND EXPLORATION OF PAM-RT ON NAS-BENCH-101

We show PAM-RT strikes a balance between exploitation and exploration on NAS-Bench-101. In Figure 9, we show that on NAS-Bench-101, evolution's performance strongly correlates with the modified hill climbing rate (and the predictor's accuracy). This suggests that PAM-RT can exploit the model for improved local search. In Figure 10, we study the diversity of samples both in the population buffer and during the whole evolution process for the experiments used in Figure 11. These measurements may serve as an indicator for the degree of exploration. We observe that PAM and PAM-RT explore more than the baseline RegEvo whereas Max-Pairwise explores much less.

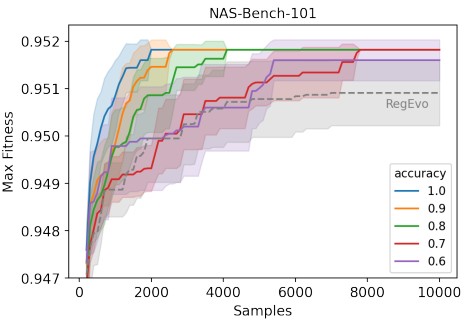 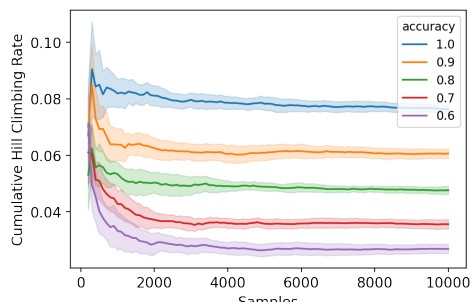

Figure 9: Strong correlation among evolution's performance, cumulative hill climbing rate, and the predictor's accuracy on NAS-Bench-101. **Left**: Evolution curves for noisy oracles of different accuracies. **Right**: Cumulative hill climbing rates in the corresponding experiments.

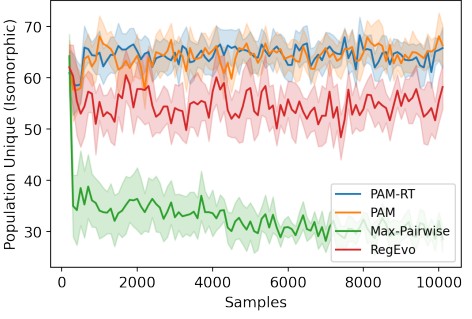 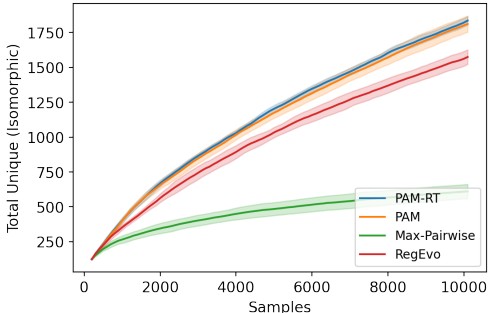

Figure 10: A followup study for Figure 11 on NAS-Bench-101 using the perfect oracle. **Left**: Number of unique candidates (by isomorphism Ying et al. (2019)) in the population buffer. **Right**: Number of total unique candidates (by isomorphism).

## 6.4 PAM-RT IS AN EFFECTIVE MUTATION STRATEGY

We show that PAM-RT is an effective way to combine binary predictor with evolution compared to other mutation strategies. In Figure 11 (left), we compare PAM, PAM-RT, and Max-Pairwise on NAS-Bench-101 using an oracle model. We observe that while all three methods converge to the same max reward value, PAM-RT reaches that value faster. In Figure 11 (right), we show the cumulative hill climbing rate for these methods along with the regularized evolution baseline. The cumulative hill climbing rate at step $N$ is defined as the average of the observed one-sample hill climbing rate at all steps up to $N$. PAM-RT and PAM both have higher rate than regularized evolution, and PAM-RT has slightly higher rate than PAM. This agrees with the analysis in Section 3.3. In Figure 9 (Appendix), we compare the cumulative hill climbing rate of PAM-RT for different noisy oracles and show that evolution's performance strongly correlates with this rate on NAS-Bench-101. Note that Max-Pairwise has an overall lower cumulative hill climb rate even compared to the regularized evolution baseline. One hypothesis is that selecting the max from a list of candidates exploits the model more in the early phase of evolution, making subsequent local improvements harder.

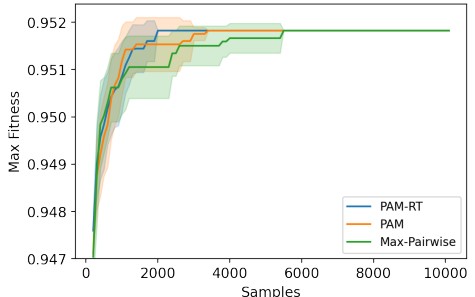 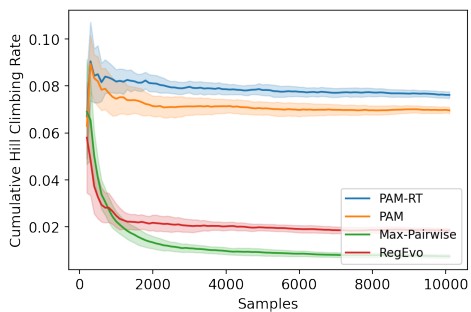

Figure 11: Comparison of three mutation strategies on NAS-Bench-101. **Left**: Fitness curves using the perfect oracle showing PAM-RT with the quickest convergence. **Right**: PAM-RT has the highest cumulative hill climbing rate compared to other strategies.

We also show PAM-RT is robust when the model is not perfect. In Figure 12a, we show that PAM-RT is similar to or better than Max-Pairwise when using noisy oracles of different accuracies. In Figure 12b, we show results with learned models and compare them with regularized evolution. Both PAM-RT and Max-Pairwise are better than the baseline, but PAM-RT reaches the best performance faster than Max-Pairwise. We emphasize that the same model architecture and training logic are used in both cases.

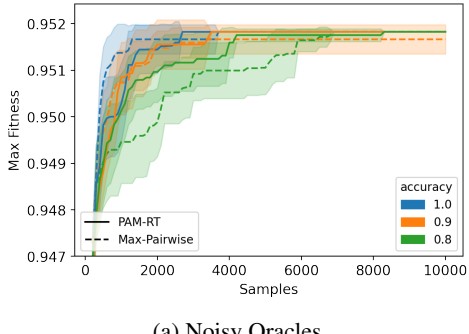 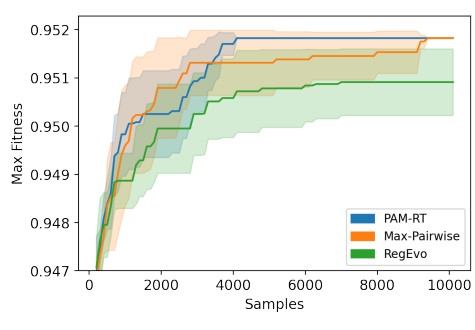

(a) Noisy Oracles

(b) Learned Models

Figure 12: Comparison of PAM-RT and Max-Pairwise on NAS-Bench-101. **Left**: Fitness curves showing PAM-RT (solid line) is more robust to predictor accuracy using a noisy oracle compared to Max-Pairwise (dashed line). **Right**: Using an online learned predictor and regularized evolution, PAM-RT is still better than Max-Pairwise.