# OpenReview forum: "Guided Evolution with Binary Discriminators for ML Program Search"
_ICLR.cc/2024/Conference — Submitted to ICLR 2024_

### Official Review · Reviewer_Mfci · 2023-10-26

**Soundness:** 4 excellent
**Presentation:** 3 good
**Contribution:** 3 good
**Rating:** 8
**Confidence:** 4

**Summary:**

Authors propose a method to accelerate a particular kind of population-based (PB) optimization algorithms, specifically, those aimed at evolving (i.e. optimizing) candidate solutions represented by graphs (or trees). These PB algorithms can be used to optimize (or "discover") DNN architectures (NAS search), RL functions, tackle symbolic regression problems, among other AutoML tasks. Their algorithm works by bypassing a significant portion of costly evaluations of individuals being evolved by using a GNN as a predictor of potential performance. Rather than using their GNN for predicting a pool of offspring to replace the current population -as recent previous approaches have proposed- they designed their predictor to be used in a 1-vs-1 basis, and then use it to compare a selected parent vs the offspring it generated, in an attempt to determine if replacing the parent with the generated offspring would be a good guess. This new approach effectively results in a new variant of regularized evolution (RegEvo). Authors provide a comprehensive battery of experimental tests and ablation studies to report the performance of their method. Results are impressive.

**Strengths:**

- Draft is very well written. Language is clear at all times, the structure and sequence make complete sense, and figures and plots are in general easy to understand.
- The proposed concept is not only very innovative, but also very *sane*: rather than using a predictor to rank a range of generated offspring, just better use it to decide which would be the best among two options. It makes sense because it sounds to be less error-prone. And as their ablation study clarifies, it ultimately works by deciding not which one of two options is better, but which one "might be less worse".
- The extended array of AutoML problems in which their approach can be used it's also a very positive point to highlight. The paper may also even work as a mini-reference that covers several AutoML areas, such that future works could refer to it for established algorithms and datasets for each one of those.
- The experimental assessment is also very well thought and done. When reading the draft, almost all questions I was coming up with were immediately answered in the experimental results sections.

Overall, the idea is clever, the results are impressive, and the paper is very well done. I have no doubts in recommending for its acceptance.

**Weaknesses:**

Having talking about the strengths of the paper, now I have to say it that I actually love it and hate it at the same time, or maybe I hate to love it.

We must understand that these kind of approaches are obfuscating an already quite obscure area: deep learning. DNN are essentially black boxes; efforts to disentangle their inner workings haven't yielded very tangible fruits yet. AutoML and NAS add another layer of obscurity. Even Genetic Programming, when used for serious task and not just toy problems, result in very large and incomprehensible syntax trees. And now here, by using NN-based predictors, authors are putting a black box inside another black box. How many more layers of invisibility are we willing to put on in the name of performance gains?

If we are willing to accept losing control and understanding of all these methods, accept that in order to reach new heights we must renounce to understand in detail their inner workings, then, by all means, go ahead. Otherwise, it might be wise to take a step backwards and hold off to certain technical developments.

I'm more on the first side: the simplicity, intuitiveness and overall results of the proposed approach, can't help but to entice me to acknowledge it.

**Questions:**

The work and paper is very well done, I just have a few doubts/suggestions that I'd like authors to clarify in their draft:

- I don't know why I miss the crucial part that details how real fitness assessments are balanced against usage of predictions as generations elapse. I don't know if this can be found in the supplementary material, but if so, it'd be a good idea to bring it forward to the main body of the paper; if it is in the main body, then try to make more emphasis on it, because somehow I miss it. A plot would be nice also, to really get a grasp of that proportion of real fitness calculations-vs-predictions as time progresses.

- Performance plots are expressed in performance vs samples. I understand why it is done that way, but in evolutionary computation/PB-methods research is far more common to express performance vs generations or epochs. It would be nice if authors could add a short paragraph or few sentences explaining how this "samples" concept is actually understood, how it relates to the regular concept of generations elapsed. (required samples processed by the whole population, etc.)

- Surrogate evaluation has been an active area of research in Genetic Algorithms (GA). You could probably talk a bit about this subject in your Related Works sections, specially considering that although surrogate methods have been successful for GAs, they are rarely used in Genetic Programming (GP), or at least to the best of my knowledge there is none of it -some proposed approaches evaluate on few samples, but as far as I am aware of, your method would be the very first true-surrogate evaluation approach for GP. If you research a bit and confirm this suspicion of mine, you should definitely highlight that in your draft.

---

### Official Review · Reviewer_qo6q · 2023-10-30

**Soundness:** 1 poor
**Presentation:** 3 good
**Contribution:** 1 poor
**Rating:** 3
**Confidence:** 4

**Summary:**

This paper proposed to evolve ML programs, with performance predictors and the predictor-based adaptive mutation. Some experiments have been conducted for verification. I have reviewed this paper in this year's NeurIPS. The authors have changed some inexact descriptions while the technical flaw remains.

**Strengths:**

Evolving ML program in a large space

**Weaknesses:**

As summarized by AC from NeurIPS, the technical contributions are limited. In addition, The paper does not compare with other state-of-the-art methods for evolutionary AutoML, such as RL or generative models.

**Questions:**

See Above

---

### Official Review · Reviewer_3xvf · 2023-11-01

**Soundness:** 3 good
**Presentation:** 3 good
**Contribution:** 2 fair
**Rating:** 5
**Confidence:** 4

**Summary:**

The paper presents a method for improving mutation in evolution within AutoML tasks. It utilizes a binary discriminator trained online to select better programs without evaluations, thus speeding up the evolution process. This method accommodates various ML components and demonstrates substantial speedups, including a 3.7x improvement for ML optimizers and a 4x boost for RL loss functions, by combining a directed acyclic graph representation with GNNs and use it to guide mutations. This approach shows potential in enhancing AutoML efficiency and performance.

**Strengths:**

The idea of guided mutation is interesting and well-inspired. It is great to see that this method uses a general approach that work with many different AutoML tasks, where the genotypes can be represented by DAGs. This general approach tries to predict the performance of a solution without evaluation, and thus filter out bad solutions in mutation to hopefully speed up evolution. While all the components are from existing work, the method itself is novel given to its genrality and application on this specific topic.

The method is well-described with a few alternatives serving as comparison methods.

The experiments are comprehensive and well-explained. The results show effectiveness of the proposed method in achieving better results using the same sample size as baselines. The ablation studies are also well-designed and further adds to the conclusion.

**Weaknesses:**

The main issue of the paper is that the author claims to improve the efficiency of evolution by reducing the sample size needed for evaluation, but completely ignores the computation cost of the proposed method itself. While we do see that the method seems to slightly reduce the sample size, it is unclear if there is really a speed-up in evolution. There are many factors that will affect the actual time, such the frequency of online training and the size of GNN, the average inference speed of each prediction, etc. The authors should definitely take into account these computation cost, and compensate the baseline for more samples to run for fair comparisons.

Besides, some other specific weaknesses and questions:
1. Nguyen is a simple benchmark, and regularized evolution isn't the best method on this benchmark. The authors should try to at least demonstrate the average performance of all problems, or switch to a more recent benchmark (e.g., SRBench) for more convincing results.
2. In addition to regularized tournament, the authors should compare to other selection methods to show generality of their mutation method.
3. On Hero and AutoRL, the method doesn't show significant improvement especially when the performance seems to start to converge. Are those results actually the best converging results? As what we really care is the efficiency to achieve the BEST result, better performance in early stages does not matter too much, so I'm not sure if the method is actually useful in anyway.
4. In figure 4, the oracle experiments are great. I wonder what's the accuracy of the predictor models that are online trained? It would be nice to include an analysis to further validate the method.
5. Does the mutation rate and the actual mutation method affect the performance of the proposed strategy?

**Questions:**

See weakness.

---

> ### Author Response · Authors · 2023-11-21
>
> We agree that the computation cost of the method itself needs to be taken into account. For simple tasks such as symbolic regression, the method will dominate the compute cost because evaluation is very cheap. The symbolic regression benchmark was more used as a cheap toy task to see if this method could speed up evolution. For more expensive tasks like Hero and AutoRL, the evaluation still dominates the computation. First we note that in Figure 3, for AutoRL the x-axis is CPU hours and includes the model prediction time (although not training time which is done on a separate server in parallel).
>
> Second, there are two components to the cost, wall clock time and total compute. Hero and AutoRL use an asynchronous training setup where the binary model is trained asynchronously with evolution so this will not add to the overall wall clock time. For compute there is another GPU server to train the model so this adds an extra GPU to the existing 20 evaluation worker GPUs for Hero and an extra GPU to the 100 existing CPU workers for AutoRL. The number of workers is listed in Table 1.  For Hero, this is a minimal increase (5%) in compute cost. For AutoRL, this is a larger increase. However we note that as you increase the number of workers as is common for running larger evolution experiments, the cost to train the model stays the same.
>
> 1. We chose the hardest problems in Nguyen to test but we can add an aggregate metric for all problems.
> 2. Regularized evolution is a simple and popular method for AutoML. For the mutation method, we do compare Max-List with PAM-RT and PAM (Figure 10 and 11). We are curious what other selection methods you have in mind.
> 3. Figure 3 still shows better final performance after convergence of our method vs vanilla regularized evolution.
> 4. The accuracy of the predictor models varies during online training but falls between 75-90%. We keep track of this metric during training and can add it to the Appendix.
> 5. The mutation method does affect the performance as shown in Section 6.4.

---

### Meta-Review · Area_Chair_ywv2 · 2023-12-09

**Metareview:**

This paper presents an approach to evolve ML programs, with performance predictors and the predictor-based adaptive mutation. The idea seems interesting and works with different AutoML tasks where the genotype can be represented by DAGs. However, the reviewers have pointed out that the current evaluation ignores the computation  cost of the proposed method itself and that it lacks a more comprehensive comparison with the state-of-the-art such as RL or generative models. Unfortunately, the authors have not addressed these issues in the rebuttal. Hence, I recommend rejection.

**Justification For Why Not Higher Score:**

Critical issues not addressed in the rebuttal.

**Justification For Why Not Lower Score:**

N/A

---

### Decision · Program_Chairs · 2024-01-16

Reject